# Densification, Tailored Microstructure, and Mechanical Properties of Selective Laser Melted Ti–6Al–4V Alloy via Annealing Heat Treatment

**DOI:** 10.3390/mi13020331

**Published:** 2022-02-19

**Authors:** Di Wang, Han Wang, Xiaojun Chen, Yang Liu, Dong Lu, Xinyu Liu, Changjun Han

**Affiliations:** 1School of Mechanical and Automotive Engineering, South China University of Technology, Guangzhou 510641, China; mewdlaser@scut.edu.cn (D.W.); 202020100649@mail.scut.edu.cn (H.W.); xjchan001@163.com (X.C.); 2Laboratory of Impact and Safety Engineering, Ministry of Education, Ningbo University, Ningbo 315211, China; 3State Key Laboratory of Vanadium and Titanium Resources Comprehensive Utilization, Pangang Group Research Institute Co., Ltd., Panzhihua 617000, China; ludong_1786@163.com (D.L.); cgvermouth2022@163.com (X.L.); 4Sichuan Advanced Metal Material Additive Manufacturing Engineering Technology Research Center, Chengdu Advanced Metal Materials Industry Technology Research Institute Co., Ltd., Chengdu 610300, China

**Keywords:** additive manufacturing, selective laser melting, laser powder bed fusion, Ti–6Al–4V, heat treatment, annealing

## Abstract

This work investigated the influence of process parameters on the densification, microstructure, and mechanical properties of a Ti–6Al–4V alloy printed by selective laser melting (SLM), followed by annealing heat treatment. In particular, the evolution mechanisms of the microstructure and mechanical properties of the printed alloy with respect to the annealing temperature near the β phase transition temperature were investigated. The process parameter optimization of SLM can lead to the densification of the printed Ti–6Al–4V alloy with a relative density of 99.51%, accompanied by an ultimate tensile strength of 1204 MPa and elongation of 7.8%. The results show that the microstructure can be tailored by altering the scanning speed and annealing temperature. The SLM-printed Ti–6Al–4V alloy contains epitaxial growth β columnar grains and internal acicular martensitic α′ grains, and the width of the β columnar grain decreases with an increase in the scanning speed. Comparatively, the printed alloy after annealing in the range of 750–1050 °C obtains the microstructure consisting of α + β dual phases. In particular, network and Widmanstätten structures are formed at the annealing temperatures of 850 °C and 1050 °C, respectively. The maximum elongation of 14% can be achieved at the annealing temperature of 950 °C, which was 79% higher than that of as-printed samples. Meanwhile, an ultimate tensile strength larger than 1000 MPa can be maintained, which still meets the application requirements of the forged Ti–6Al–4V alloy.

## 1. Introduction

Ti–6Al–4V alloy has been widely used in the aerospace, energy, biomedical, and automotive sectors [1,2] due to its high strength, low density, high fracture toughness, excellent corrosion resistance, and good biocompatibility [3]. Metal additive manufacturing (AM) has been advancing in the fabrication of geometrically complex metal products, typically including selective laser melting (SLM), directed energy deposition, metal binder jetting, and sheet lamination [4,5,6]. SLM has been widely applied to manufacture complex titanium parts with short lead time, great design freedom, and comparable product performance to forged counterparts [7], such as aircraft brackets [8], cervical fusion cages [9], bone implants [10], and partial denture clasps [11].

A considerable number of research works have been conducted on the fabrication of the Ti–6Al–4V alloy via SLM. Process parameters have great influence on the relative density, microstructure, and mechanical properties of SLM-printed Ti–6Al–4V alloy. For instance, Sun et al. [12] explored the influence of laser power and scanning speed on the relative density of SLM-printed Ti–6Al–4V parts. With an increase in the laser power and a decrease in the scanning speed, the relative density of the printed parts could increase to more than 99%. Yang et al. [13] found that the microstructure of SLM-printed Ti–6Al–4V samples was composed of a typical hierarchical martensite structure with a high density of dislocations and twins, including primary, secondary, tertiary, and quaternary α′ martensite in β columnar grains. The process parameters can affect the temperature and cooling rate of the melt pools, thus affecting the microstructure and mechanical performance of the printed Ti–6Al–4V parts. Wang et al. [14] established the relationship among process parameters, microstructure evolution, and mechanical properties. With an increase in the scanning speed to 1150 mm/s, the elongation could reach the maximum value of 7.8%. The synthetic effects of the grain refinement of α (α′) martensite and the nano-β particle resulted in the improvement of the elongation. The SLM-printed Ti–6Al–4V samples usually possess higher strength and microhardness than cast or forged counterparts [15,16,17,18]. If nonoptimized process parameters are applied, manufacturing defects such as balling, cracks, and porosity are prone to appear, which are detrimental to the mechanical performance of the parts [19,20].

Post-treatments by annealing and hot isostatic pressing (HIP) are commonly applied to SLM-printed Ti–6Al–4V alloy to improve its elongation by transforming the α′ martensite phase into a mixture of α and β phases [21,22,23,24,25,26,27]. Wang et al. [23] conducted annealing on the printed samples at 840 °C (below β phase transition temperature) and found that the maximum elongation increased from 5.79% to 10.28%, while the fracture type changed from quasi-cleavage to ductile fracture. Jamshidi et al. [27] performed HIP for the printed samples along the horizontal and vertical orientations at 930 °C and 100 MPa for 4 h. The results showed that the ductility was improved 2.1- and 2.9-fold in the vertical and horizontal orientations, respectively. After the post-treatments, the SLM-printed Ti–6Al–4V alloy could obtain improved plasticity but decreased mechanical strength.

Limited research has been systematically performed on the effects of process optimization and heat treatment on the microstructure and properties of SLM-printed Ti–6Al–4V alloy. Additionally, most research focused on the investigation of the microstructure and property evolution of the SLM-printed Ti–6Al–4V alloy heat-treated below the β phase transition temperature. This study aimed to determine the densification, tailored microstructure, and mechanical properties of the SLM-printed Ti–6Al–4V alloy through annealing heat treatment. In particular, the evolution mechanisms of the microstructure and mechanical properties of the printed alloy with respect to the annealing temperature near the β phase transition temperature were investigated. The process parameter optimization of SLM was conducted to obtain a high degree of densification for the printed alloy. The influences of scanning speed and annealing temperature on the microstructure and mechanical properties of the alloy were investigated and analyzed.

## 2. Materials and Methods

### 2.1. Materials

An atomized Ti–6Al–4V alloy powder (AP&C company, Boisbriand, QC, Canada) with an average particle size of 33 μm was used. The material composition is shown in Table 1. The loose density of the powder was 2.45 g/cm^3^. Figure 1 shows the morphology of the powder and its particle size distribution. The powder particles were almost entirely spherical, and the particle size distribution was 15–45 μm.

### 2.2. SLM Process and Heat Treatment

A Dimetal-100 SLM equipment (Laseradd Technology Co., Ltd., Guangzhou, Guangdong, China) was utilized to print the Ti–6Al–4V alloy powder. The process optimization was conducted, and the parameter variables are shown in Table 2. The scanning strategy was bidirectional orthogonal scanning with a scanning starting angle of 145°. The cubic samples with dimensions of 8 mm × 8 mm × 8 mm and tensile samples were vertically printed, respectively, using various laser power, scanning speed, and hatch space values. All the samples were printed at the same position, i.e., central area of the substrate.

Annealing heat treatment was applied to the Ti–6Al–4V samples printed using optimized process parameters. It was reported that the annealing heat treatment for the SLM-printed Ti–6Al–4V alloy was mostly conducted at the temperature of 600–750 °C for 2 h, and the favorable temperature range was 800–900 °C for other heat treatments (except annealing) in the same timespan of 2 h [28]. In addition, the Ti–6Al–4V alloy annealed near to the β phase transition temperature (995 °C) could obtain the most improved mechanical properties [29]. Therefore, the annealing temperatures were set at 750 °C, 850 °C, 950 °C, and 1050 °C respectively. The samples were then held in the furnace for 2 h and then cooled. The procedure of the annealing process is shown in Figure 2.

### 2.3. Characterizations

After printing, the relative density of the samples was measured to determine the optimal process parameters. The relative density of the printed Ti–6Al–4V samples was measured by an OHAUS PX124ZH electronic analytical balance (OHAUS Corporation, Parsippany, NJ, USA) according to the Archimedes drainage method. The samples were ground, polished, and etched with Kroll’s agent (H_2_O/HF/HNO_3_ = 1:3:50 mL) for 20 s for microstructure characterization. The microstructure of the samples was observed with a Leica inverted optical microscope (Leica Microsystems GmbH, Wetzlar, Germany). The phase identification of the printed samples before and after annealing was conducted by a D8 ADVANCE X-ray diffractometer with copper target X-ray (Bruker AXS GmbH, Karlsruhe, Germany), with a scanning speed of 4 °/min, scan angle of 20–80°, and voltage value of 40 kV. The fracture morphology of the samples was observed by a Quanta 250 scanning electron microscope (FEI Company, Hillsboro, Oregon, USA).

### 2.4. Mechanical Testing

Tensile samples were printed according to ASTM-E8 (65 mm height, 25 mm gauge length, 2 mm thickness, and 5 mm width). Three samples were tested for each scanning speed and annealing temperature. The tensile tests at room temperature were carried out on a CMT5504 electronic universal testing machine (Zhuhai SUST Electrical Equipment Co., Ltd., Zhuhai, Guangdong, China) with an NCS electronic extensometer (NCS Testing Technology Co., Ltd., Beijing, China) and a speed of 0.5 mm/min.

## 3. Results and Discussion

### 3.1. Relative Density

Figure 3 shows the relative density of the SLM-printed samples with various scanning speed and laser power values when the hatch space was 0.07 mm, and the layer thickness was 0.03 mm. Under a low laser power (140–150 W), the relative density increased from 98.3% to 99.04% with the scanning speed increasing from 700 mm/s to 900 mm/s, and then decreased to 97.4% when the scanning speed was larger than 900 mm/s (Figure 3a). A low laser power is prone to producing lack of fusion pores [30]. At low scanning speeds, the laser beam can continuously heat the melt pools, resulting in greater laser energy to the Ti–6Al–4V powder particles and more unstable molten pool flow [31]. It is easy to trap gas into the melt pools to form micropores during their solidification [32]. Meanwhile, serious sputtering occurs, and splashed metal particles fall back to the surface of powder bed to form metal spheres. The reason for the decrease in the relative density is that the accumulation of the spheres leads to the generation of inclusions and pores [33].

Comparatively, under a relatively high laser power (160–180 W) (Figure 3b), the relative density could reach the highest values at a scanning speed of 1300 mm/s. A high laser power can result in a large depth of the powder layer penetrated by the laser beam, which improves the fluidity of the melt pool [34]. At high scanning speeds, a large solidification shrinkage of the melt pools tends to occur, resulting in a poor multi-track overlap and large gap between the tracks. The increase in the gap leads to the increase in layer thickness in the track gap after powder spreading. Therefore, the effective energy density is reduced, which promotes the formation of pores and reduces the relative density of the printed samples. The highest relative density of 99.51% could be achieved with a laser power of 170 W and a scanning speed of 1300 mm/s for the SLM-printed Ti–6Al–4V alloy.

Figure 4 shows the variation of the relative density of the SLM-printed samples with hatch space when the laser power was 170 W, the scanning speed was 1300 mm/s, and the layer thickness was 0.03 mm. When the hatch space increased from 0.06 mm to 0.07 mm and from 0.07 mm to 0.1 mm, the relative density of the sample increased from 99.21% to 99.5% and then gradually decreased to 98.5%, respectively. When the laser power and scanning speed were kept constant, the laser input energy was constant, and the melt pool width remained stable. A large hatch space resulted in a quite small overlap rate between adjacent melt pools, which is conducive to the formation of porosity. Comparatively, the decrease in the hatch space increased the overlap rate and reduced the heating time interval between the adjacent melt pools, resulting in a sufficient metal flow within the melt pools and resultant high relative density [35].

### 3.2. Microstructure

Figure 5 exhibits the microstructure of the SLM-printed Ti–6Al–4V samples manufactured at different scanning speeds under a laser power of 170 W, hatch space of 0.07 mm, and layer thickness of 0.03 mm. The microstructure of the samples with different scanning speeds was mainly composed of coarse epitaxial columnar grains that grew along the building direction. In the SLM process, the melt pool temperature is generally higher than that of the β phase generation. The ultrahigh cooling rates (up to 10^6^ K/s) suppresses the transformation from the β phase into the α phase, and martensitic transformation occurs to form fine acicular α′ grains [36]. The primary β grains were filled with fine acicular α′ martensite that grew toward 45° upward with the building direction. When the scanning speed ranged from 900 mm/s to 1100 mm/s, the average width of the primary β grain was about 200 μm. However, the further increase in the scanning speed from 1200 mm/s to 1300 mm/s resulted in a decrease in the average width of the primary β grain to 150 μm. In addition, the scanning speed had a great impact on the porosity of the samples, which is consistent with the results shown in Figure 3.

Figure 6 shows the X-ray Diffraction (XRD) pattern of the SLM-printed Ti–6Al–4V samples with different scanning speeds. Both α and α′ phases possessed hexagonal close-packed (hcp) structures. The diffraction angle of the strongest peak of the samples was shifted to a large value, as compared to the standard diffraction angle of 40.251°. This peak shift confirmed the formation of the martensite α′ phase. The increase in scanning speed led to the larger shift of the α′ peak, due to the increase in the cooling rate of the melt pool [37]. The cooling rate increase reduced the β precipitated phase but increased the content of V and Al in the acicular α′ phase, which decreased the lattice size of the α′ phase and the augmentation of its diffraction angle [14]. In the spectrum, the diffraction peak of the β phase was not obvious because of its low content caused by its transformation into the α′ phase during the cooling process.

### 3.3. Mechanical Properties

Figure 7 shows the mechanical properties of the SLM-printed Ti–6Al–4V samples with different scanning speeds. The scanning speed had a significant effect on the elongation but not on the tensile strength. The tensile strength of the samples was in the range of 1200–1265 MPa, and the maximum value could be obtained when the laser power was 170 W and the scanning speed was 900 mm/s. However, the elongation firstly increased from 5.5% to 7.8% and then decreased with an increase in the scanning speed. The maximum elongation could be obtained at a scanning speed of 1300 mm/s. Since the as-printed Ti–6Al–4V samples contained complete martensite structures and fine grains, their tensile strength was much larger than the minimum strength requirements for forged Ti–6Al–4V specified in the standard ASTM F1472-14, but the elongation was lower due to the brittleness of the martensite and large residual stress.

The scanning speed could influence the amount of the precipitated β phase and the size of the α′ martensite in the SLM-printed samples. The β phase distributed at the boundary of the acicular α′ martensite phase had higher strength, and the decrease in elongation was mainly due to the dislocation locking by β during tensile stress [38], which hindered the movement of dislocations between the α phases. Therefore, with the increase in the scanning speed, the β phase decreased and the elongation of the samples increased. On the other hand, the reduction in the slip length of the α phase may have resulted in the increase in elongation [39]. The slip length of the α phase could be approximately equal to the width of the acicular α′ martensite. Upon increasing the scanning speed, the α′ martensite was refined, thereby reducing the slip length of the α phase and increasing the elongation.

Figure 8 presents the representative tensile fracture morphology of the SLM-printed samples under a scanning speed of 900 mm/s and 1300 mm/s. When the scanning speed was 900 mm/s, the fracture surface was mainly composed of flat cleavage steps, and also contained shallow dimples, indicating brittle fracture. The edge of the sample was exposed with more pore defects and nonmolten powder particles. These defects came from the insufficient laser energy input during the SLM process and were strongly related to the improper scanning speed. During the tensile testing, the pores led to the initiation of cracks, causing premature failure of the sample. Under a scanning speed of 1300 mm/s, the sample showed few internal defects and shearing surfaces. The higher scanning speed reduced the size of the crystal grains, which exhibited higher ductility under tension [12].

### 3.4. Effect of Annealing Temperature on Microstructure

The Ti–6Al–4V alloy samples before annealing were printed with a laser power of 170 W, a scanning speed of 1300 mm/s, and a hatch space of 0.07 mm. Figure 9 presents the XRD pattern of SLM-printed Ti–6Al–4V alloy samples after annealing at different temperatures. The results showed similar α and β diffraction peaks of the samples at different annealing temperatures. However, the diffraction peak of the β phase was weak, indicating the low volume fraction of the β phase. The full width at half maximum (FWHM) of the α/α′ peak (2θ = 40.2°) is listed in Table 3. Compared to the SLM-printed sample, the FWHM of the heat-treated samples significantly decreased, elucidating that the residual stress within the printed samples was significantly eliminated [40].

The microstructure of the SLM-printed samples after annealing at different temperatures is shown in Figure 10. When the annealing temperature of 750 °C was applied, the microstructure of the sample changed significantly as compared with that of the as-printed sample (Figure 10a,b). The primary β columnar grains in the annealed sample still existed, while the acicular α′ martensite in the columnar grains transformed into a mixed α + β phase. The annealing temperature of 750 °C could only drive the partial decomposition of the α′ martensite; thus, the α phase in the structure maintained the acicular shape [41]. At the annealing temperature of 850 °C, there was still epitaxial growth of the β columnar grains, but the β boundary became blurred and disappeared (Figure 10c,d). Compared with the annealed microstructure at 750 °C, the metastable acicular α′ martensite phase in the β columnar crystals almost decomposed, and the lath-shaped α phase increased and became coarse. After annealing at 850 °C, the microstructure consisted of the α and β phase, showing a network structure. Studies have shown that the α′ phase of Ti–6Al–4V alloy can completely decompose at temperatures above 800 °C [42].

After annealing at 950 °C, β columnar grains disappeared and the lamellar α phase could be observed (Figure 10e,f). The grains were further coarsened, and the distribution of α + β dual phase was more uniform. When the annealing temperature increased to 1050 °C, exceeding the β phase transition temperature, the microstructure was completely β phase in the heat preservation state, and the columnar structure in the as-printed sample could be completely eliminated (Figure 10g,h). Due to the slow cooling rate, the α phase in the β grain gathered to form a lath-shaped structure with the same orientation, and coarse Widmanstätten structures could be obtained [43].

### 3.5. Effect of Annealing Temperature on Mechanical Properties

Table 4 and Figure 11 show the tensile properties of the SLM-printed samples at different annealing temperatures. It can be seen that the tensile strength decreased gradually with the increase in the annealing temperature. When the annealing temperature of 750 °C was applied, the tensile strength of the sample decreased to 1094 MPa, which is 9% lower than that of the printed sample. The elongation of 7% was similar to that of the printed sample. The maximum elongation of 14% could be obtained at 950 °C, which is 79% higher than that of the as-printed sample.

After annealing at 750 °C, some brittle and hard α′ martensites decomposed into the α + β phase with relatively high ductility. However, the partial decomposition suppressed the change in the elongation of the sample but reduced its tensile strength. When the annealing temperature exceeded 800 °C, the acicular α′ martensite completely decomposed into the α and β phases, which decreased the tensile strength and gradually increased the elongation. When the annealing temperature exceeded the β phase transition temperature, the formed coarse grains and the lath-shaped α phases inside them could hinder the slip of dislocations, causing stress concentration at the interface of the α and β phases and eventually reducing the ductility.

Figure 12 shows the fracture morphology of the samples at different annealing temperatures. It can be observed that the fracture morphology of the samples annealed at 750 °C and 850 °C was similar to that of the as-printed samples, including cleavage facets and dimples. However, when the annealing temperature increased to 950 °C, the cleavage facets disappeared and dense dimples with large sizes were formed, indicating a ductile fracture. When the local stress at the phase interface exceeded the interfacial bonding force in the tensile process, micropores occurred and consumed a large amount of strain energy. Micropores lead to dimples in the aggregation process, and denser dimples indicate the better ductility. After annealing at 1050 °C, the dimple size increased, and a small number of tear ridges appeared, which reduced the elongation of the sample.

Table 5 shows the mechanical properties of Ti–6Al–4V samples after different post-treatments as compared to those reported in previous studies. It can be seen that the sample after 950 °C heat treatment exhibited a superior tensile strength and reasonable elongation. Heat treatment temperatures below 950 °C reduced the β phase [23], resulting in a higher tensile strength of the printed alloy. The samples with higher elongations were treated by HIP [26,27], which was more conducive to tailoring the microstructure and reducing the pore defects in the sample.

## 4. Conclusions

This work investigated the microstructure and mechanical properties of the SLM-printed Ti–6Al–4V alloy post-treated by annealing. The effects of process parameters on the relative density, microstructure, and mechanical properties of SLM-printed samples were studied. The effects of annealing temperature on microstructure and mechanical properties of the printed samples were further studied. The main findings are presented below.

The relative density of the SLM-printed sample was significantly affected by the scanning speed. In particular, the SLM-printed sample could obtain the highest density of 99.51% with a laser power of 170 W, a scanning speed of 1300 mm/s, a layer thickness of 0.03 mm, and a hatch space of 0.07 mm.

The microstructure of the printed sample was composed of β columnar crystals, which contained a large number of acicular α′ martensite, resulting in higher strength and lower plasticity of the sample. The width of the β columnar crystals decreased with the increase in the scanning speed, as determined by the decrease in the energy density. The maximum tensile strength of 1265 MPa was achieved at a scanning speed of 900 mm/s, while its elongation could reach the highest value of 7.8% at a scanning speed of 1300 mm/s.

The annealing temperature had a significant effect on the microstructure of the sample. After annealing, the acicular α′ martensite was decomposed into the α + β dual phase. With the increase in the annealing temperature, the tensile strength gradually decreased, while the elongation increased first and then decreased. Annealing at 950 °C could result in the highest elongation of 14%, which is 79% higher than that of the as-printed sample, without a significant reduction in the tensile strength.

## Figures and Tables

**Figure 1 micromachines-13-00331-f001:**
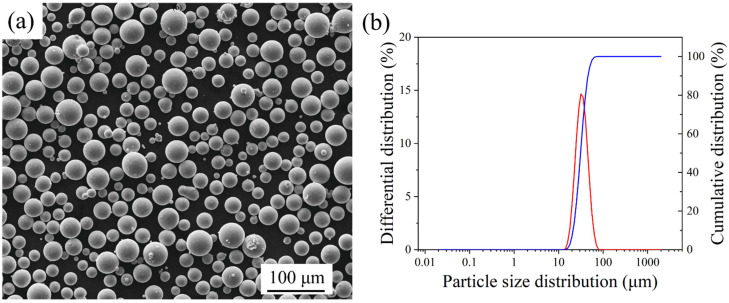
Characteristics of the Ti–6Al–4V alloy powder: (**a**) morphology; (**b**) particle size distribution.

**Figure 2 micromachines-13-00331-f002:**
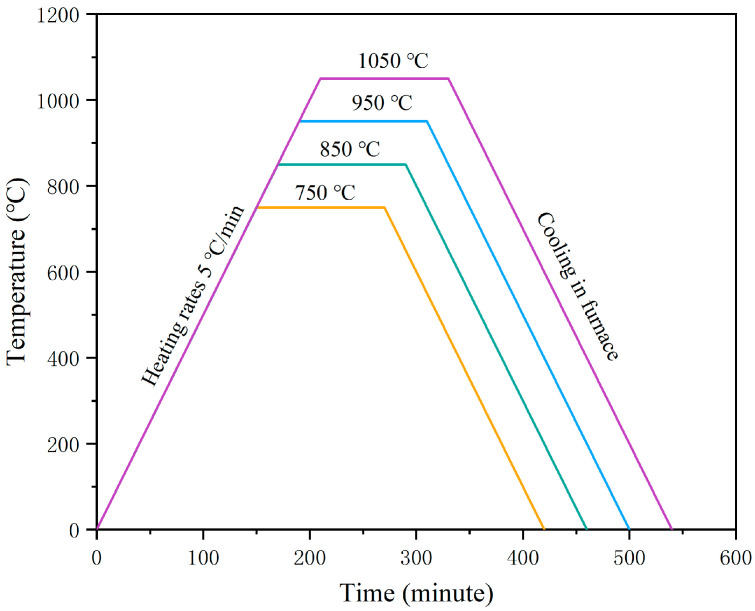
Annealing heat treatment procedures for the SLM-printed Ti–6Al–4V alloy.

**Figure 3 micromachines-13-00331-f003:**
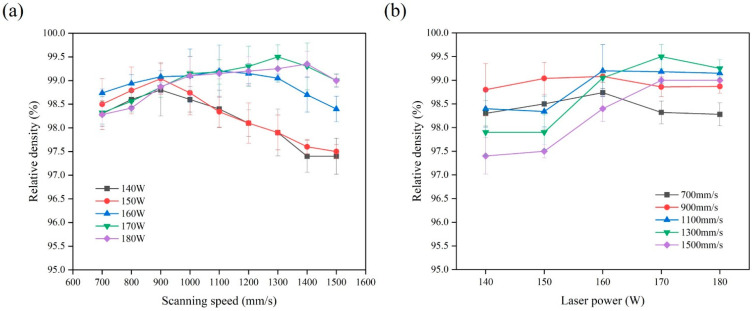
Variation trend of relative density of Ti–6Al–4V samples with various parameters: (**a**) scanning speed; (**b**) laser power.

**Figure 4 micromachines-13-00331-f004:**
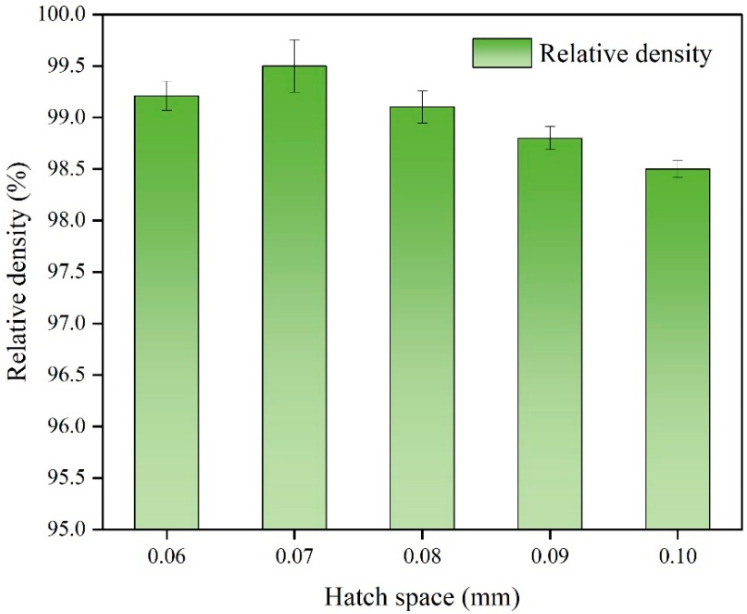
Histogram showing the influence of the hatch space on the relative density of the SLM-printed Ti–6Al–4V samples.

**Figure 5 micromachines-13-00331-f005:**
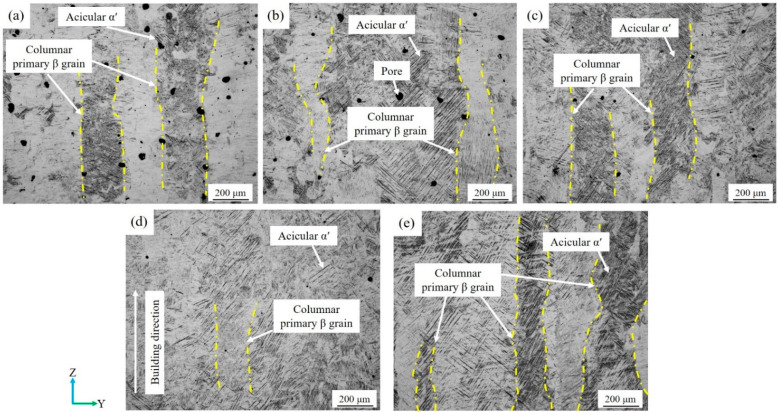
Microstructure of the SLM-printed Ti–6Al–4V samples at different scanning speeds: (**a**) 900 mm/s; (**b**) 1000 mm/s; (**c**) 1100 mm/s; (**d**) 1200 mm/s; (**e**) 1300 mm/s.

**Figure 6 micromachines-13-00331-f006:**
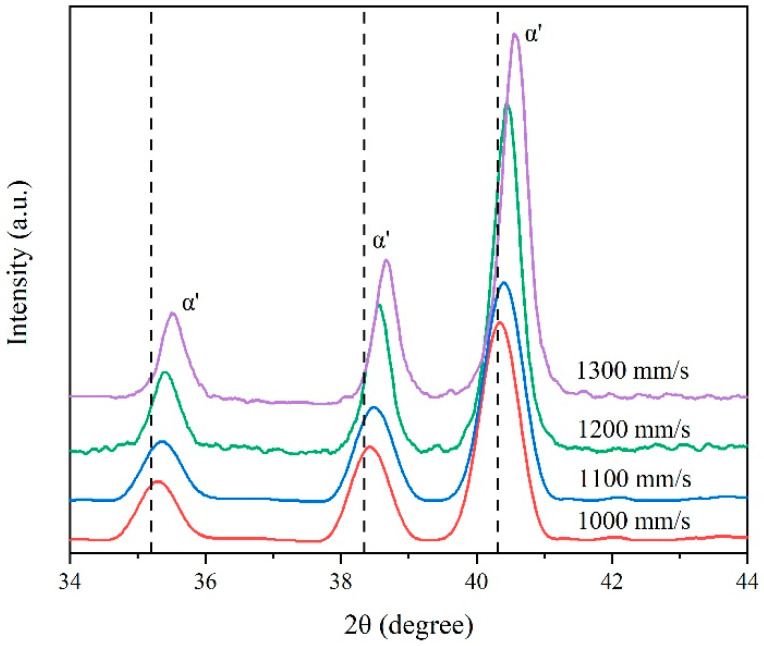
X-ray Diffraction (XRD) pattern of the SLM-printed Ti–6Al–4V samples at different scanning speeds from 34° to 44°.

**Figure 7 micromachines-13-00331-f007:**
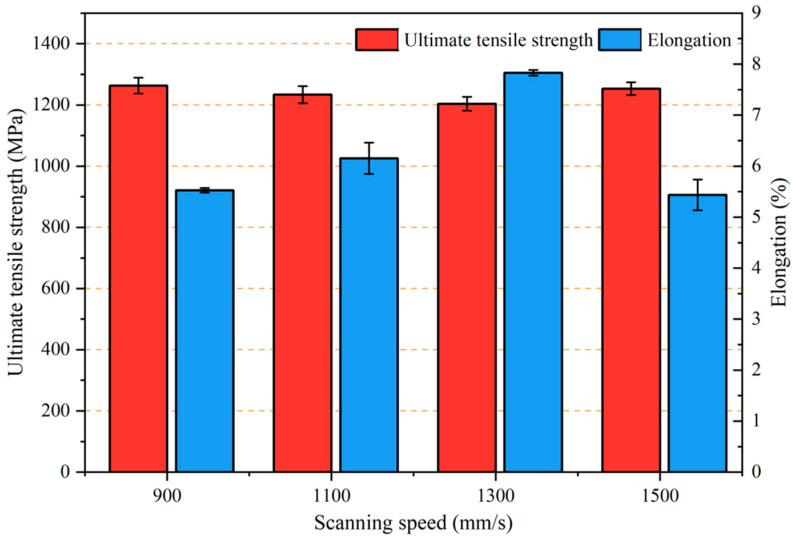
Effect of the scanning speed on the mechanical properties of the SLM-printed Ti–6Al–4V alloy.

**Figure 8 micromachines-13-00331-f008:**
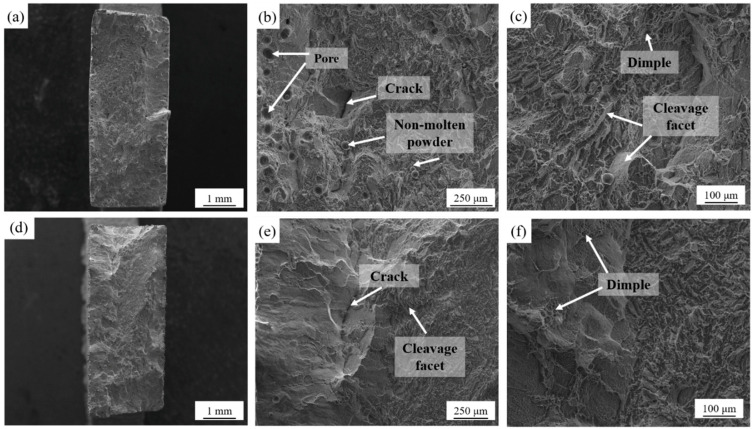
Fracture morphology of the SLM-printed Ti–6Al–4V samples at different scanning speeds: (**a**–**c**) 900 mm/s; (**d**–**f**) 1300 mm/s.

**Figure 9 micromachines-13-00331-f009:**
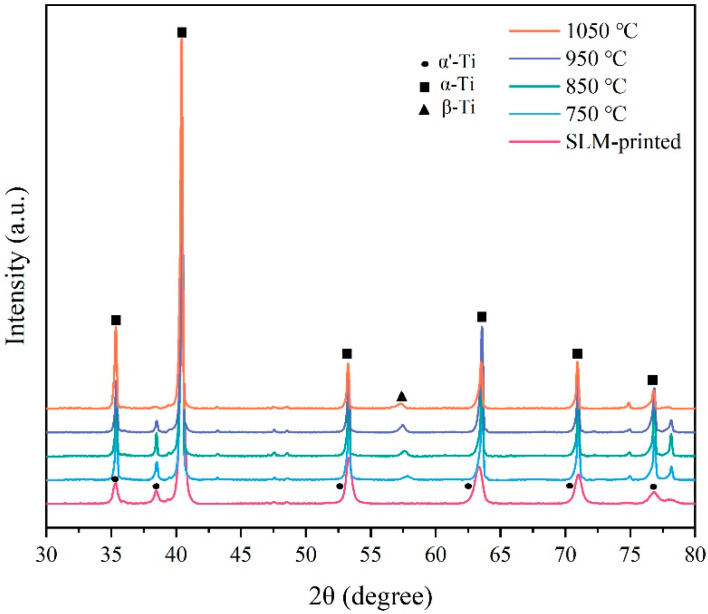
XRD pattern of the SLM-printed samples before and after annealing heat treatment.

**Figure 10 micromachines-13-00331-f010:**
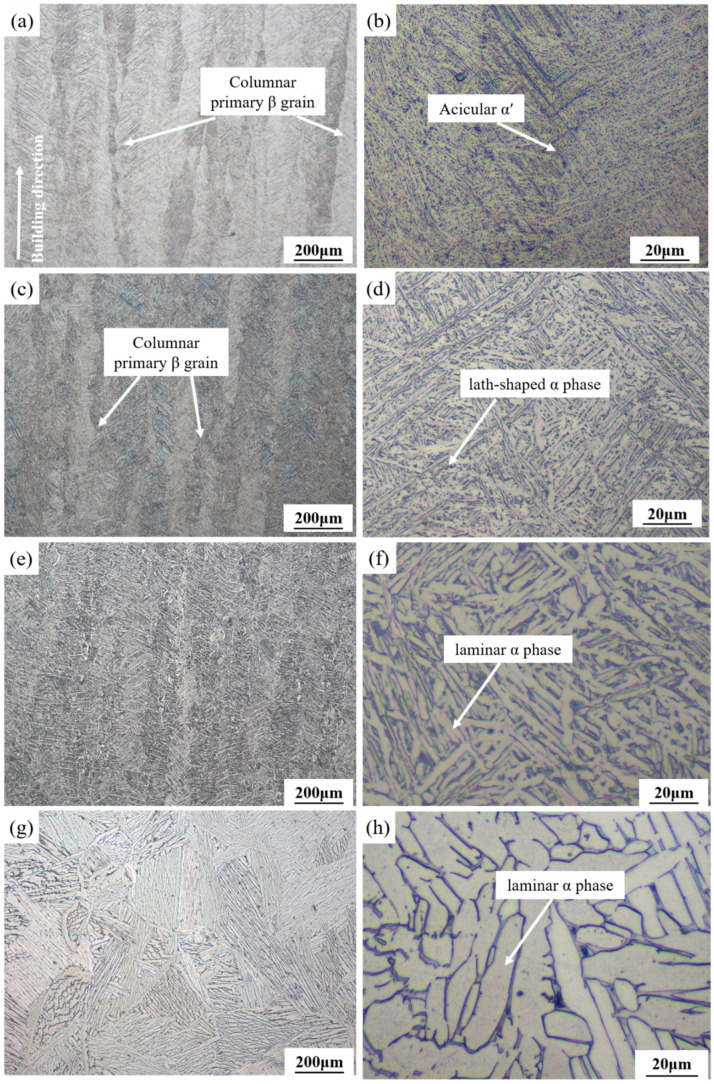
Microstructure of the printed Ti–6Al–4V samples along the build direction after annealing at different temperatures: (**a**,**b**) 750 °C; (**c**,**d**) 850 °C; (**e**,**f**) 950 °C; (**g**,**h**) 1050 °C.

**Figure 11 micromachines-13-00331-f011:**
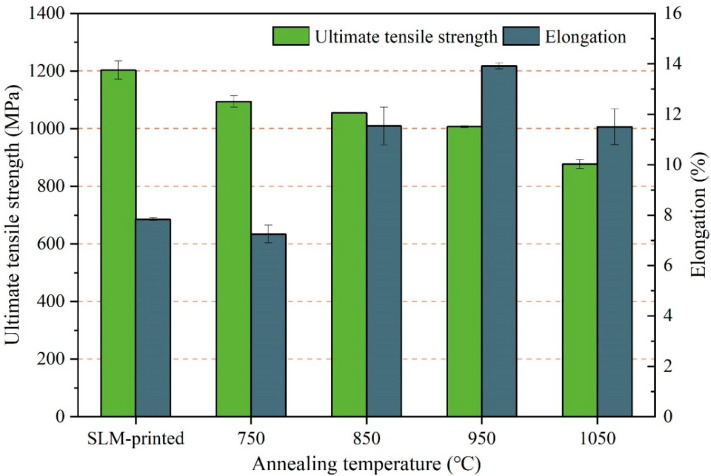
Effect of annealing temperature on the mechanical properties of the SLM-printed Ti–6Al–4V samples (laser power of 170 W, scanning speed of 1300 mm/s, and hatch space of 0.07 mm).

**Figure 12 micromachines-13-00331-f012:**
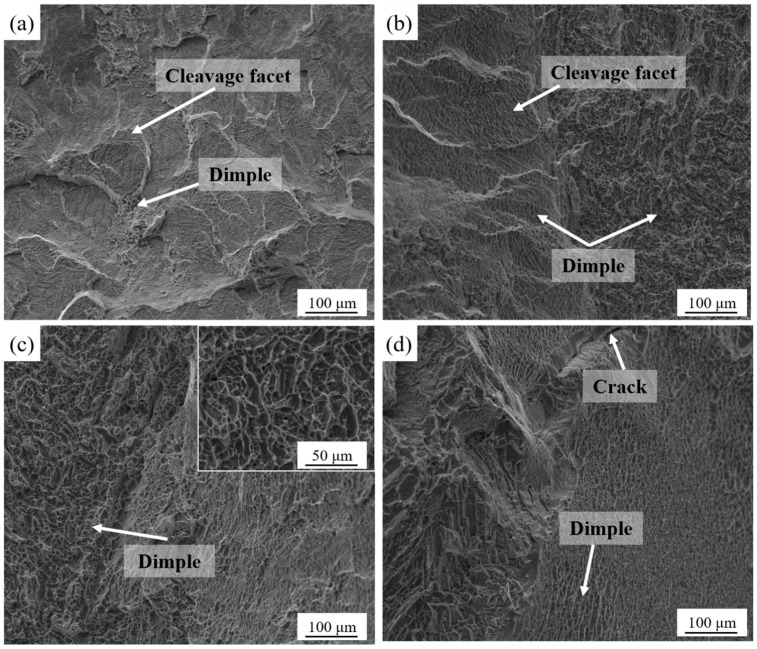
Fracture morphology of the printed samples after annealing at different temperatures: (**a**) 750 °C; (**b**) 850 °C; (**c**) 950 °C; (**d**) 1050 °C.

**Table 1 micromachines-13-00331-t001:** Chemical composition of the Ti–6Al–4V alloy powder.

Element	Ti	C	O	Ni	H	Fe	Al	V	Others
Ratio (%)	Balance	0.02	0.11	0.02	0.034	0.19	6.5	3.9	<0.1

**Table 2 micromachines-13-00331-t002:** Process parameters of selective laser melting (SLM) for printing the Ti–6Al–4V alloy powder.

Parameter	Value
Laser power (W)	140, 150, 160, 170, 180
Scanning speed (mm/s)	700, 800, 900, 1000, 1100, 1200, 1300, 1400, 1500
Hatch space (mm)	0.06, 0.07, 0.08, 0.09, 0.1
Layer thicknesses (mm)	0.03

**Table 3 micromachines-13-00331-t003:** Full width at half maximum (FWHM) of the samples calculated from the XRD patterns.

Annealing Temperature	SLM-Printed	750 °C	850 °C	950 °C	1050 °C
FWHM	0.419	0.188	0.149	0.196	0.160

**Table 4 micromachines-13-00331-t004:** Mechanical properties of the SLM-printed Ti–6Al–4V samples.

Sample	Annealing Temperature (°C)	Ultimate Tensile Strength (MPa)	Elongation (%)
1	SLM-printed	1204 ± 32	7.8 ± 0.1
2	750	1094 ± 20	7 ± 0.5
3	850	1055 ± 1	11 ± 1.5
4	950	1007 ± 3	14 ± 0.1
5	1050	877 ± 16	11 ± 1

**Table 5 micromachines-13-00331-t005:** Mechanical properties of the SLM-printed Ti–6Al–4V samples after different post treatments.

Sample Condition	Ultimate Tensile Strength (MPa)	Elongation (%)	Source
950 °C for 2 h	1007 ± 3	14 ± 0.1	This work
850 °C for 2 h	1004 ± 6	12.84 ± 1.36	[21]
890 °C for 2 h	998 ± 14	6 ± 2	[22]
840 °C for 2 h + furnace cooling to 450 °C + air cooling	1068.3 ± 26.7	10.28 ± 0.20	[23]
HIP at 920 °C and 100 MPa for 2 h	1088.5 ± 26.3	13.8 ± 1.3	[25]
HIP at 900 °C and 120 MPa for 2 h	941	19	[26]
HIP at 930 °C and 100 MPa for 4 h + wet polishing	936 ± 3.6	21.7 ± 2.3	[27]

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
