# Peer review of "Densification, Tailored Microstructure, and Mechanical Properties of Selective Laser Melted Ti–6Al–4V Alloy via Annealing Heat Treatment"

_micromachines, 2022, doi:10.3390/mi13020331_

Round 1
Reviewer 1 Report
Dear Authors,
I have really appreciated your work. It is interesting and I hope to recommend it for publication after the review I believe necessary.
Just some of the suggestions and comments:
Line 136: in this chapter the authors are reporting the results with a discussion. It is not clear the reason for the reference [33] in line 136. In fact, please, specify if the decreasing of density has been measured (and in this case report a table with data) or you expect a possible effect on density as already reported in literature [33].
Line 137: please, in this line repeat the reference to Fig 3, b.
Line 273-274: Please specify how many samples have been tested. Are the data "single-points"? Are Mean Values? Moreover, add the printing parameters details in caption of the Fig.11.
Moreover, confirm that all the samples compared have been printed with the same growing direction and plate position.
Line 274: this sentence put me in confusion: what means a “generic SLM-printed” samples? What are the printing parameters adopted for the SLM-printed samples you are comparing with different annealings in figure 11?
Are you sure that this annealing is working for all the samples produced ? Even for those printed with the worse parameters combination? Try to be more precise.
Line 295-296: Please add a discussion of the reasons for the optimisation of the fracture mode. Do you a assessment of residual stresses before and after the anneling tretaments? ? Compare the density and give a rationale for the justification of this very interesting result.
To be clearer: add a discussion for the reason you believe that the annealing at T=950°C is the best.
From the paper some point remain unclear: what are the printing-parameters adopted for the samples used for the annealing tests? Report here a table with: sample number/printing parameters/mechanical properties as-fabricated/mechanical properties after annealing.
Reviewer 2 Report
Manuscript describes the influence of SLM parameters and modes on the microstructure, phase composition and physical mechaniacl properties of Ti-6Al-4V. The manuscript is of scientific interest and of practical significance. However, there are few points should and must be corrected.
It is unclear what is the purpose of the manuscript. Introduction part contains data about SLM modes and parameters those influence on structure, physical and mechanical properties of Ti-6Al-4V, there is an information about post-treatment and its effects, etc. It is not evident, what is the novelty of the manuscript. Authors should rewrite the introduction and abstract.
Authors used powder particles with the size achieved 100 microns. At the same time the applied layer thickness during SLM was 0.03 mm, which is 30 microns. One can assume that the layer will not be completely melted during SLM. This, in its turn, may lead to the formation of unmelted areas within the bulk of produced sample, to the formation of cracks and pores. Authors should explain, why did they choose such SLM mode.
Materials and methods part does not contain information about applied modes of SEM, XRD, optical microscopy, etc. This information should be added.
There is no values' divergence in fig. 3. Authors should add it. Otherwise, it is unclear, whether the obtained results are of statistical significance.
The same is for the fig. 4.
Contrast should be increased in fig. 5, as primary beta-grains and acicular alpha'-grains are not clearly visible.
There are particles of non-molten powder within the fracture of SLM samples. Authors do not explain the presence of those particles. This information should be added.
Authors connect decreasing of FWHM in fig. 9 with the decreasing the level of residual stresses. However, residual stresses is not the only phenomena affecting the FWHM. Did authors calculated CSR sizes or dislocation density? May be this data should be provided in table 3 too?
It is unclear, why authors chose the descibed temperatures of annealing.
Discussion part is not evident within the manuscript. Authors should add the comparison of obtained results with the already published data.
Round 2
Reviewer 1 Report
All the comments have been duly addressed. Thank You.
Author Response
衷心感谢您对我们的稿件提出的意见和建议!
Reviewer 2 Report
The manuscript was significantly improved. Most part of reviewer's comments and questions was answered and clarified. However, there are still few more points required to revise.
- Description of SEM-experimental part is still absent in "2.3 Characterizations" part. This should be added.
- Authors provide such measured or calculated values as 1007.35 +- 2.35 or 1054.5 +- 0.70, etc. Is the provided accuracy of statistical importance? In reviewr's oppinion, those values could be 1005 +- 2 or 1055 +- 1.0, etc. Authors should adjust the obtained numerical data.
